# Emollient prescribing formularies in England and Wales: a cross-sectional study

Jonathan Chan, Grace Boyd, Patrick A Quinn, Matthew J Ridd

## ABSTRACT

**Objective** To identify and compare emollient formularies across all clinical commissioning groups (CCGs) and local health boards (LHBs) in England and Wales.

**Design** Formularies were retrieved via CCG/LHB websites or Google search (October 2016–February 2017). Data on structure and content were extracted, and descriptive analyses were undertaken.

**Setting** 209 English CCGs and 7 Welsh LHBs.

**Main outcome measures** Number and structure of formularies; number, type and name of emollients and bath additive recommendedandnot recommended; and any rationale given.

**Results** 102formularies were identified, which named 109 emollients and 24 bath additives. Most were structured in an 'order of preference' (63%) and/or formulation (51%) format. Creams and ointments were the most commonly recommended types of emollients, and three ointments were the most commonly recommended specific emollients (71%–79% of formularies). However, there was poor consensus over which emollient should be used first line and 4 out of 10 of the most recommended lotions and creams contained antimicrobials or urea. Patient preference (60%) and/or cost (58%) were the most common reasons given for the recommendations. Of the 82% of formularies that recommend the use of bath additives, 75% did not give any reasons for their recommendation.

**Conclusions** Emollient formularies in England and Wales vary widely in their structure, recommendations and rationale. The reasons for such inconsistencies are unclear, risk confusion and make for inequitable regional variation. There is poor justification for multiple different, conflicting formularies.

## Strengths and limitations of this study

⇒ This is the first study to compare in a systematic way all of the emollient formularies in England and Wales.

⇒ Due to the labour-intensive nature of data extraction and constant change in clinical commissioning group/local health board service configuration and formularies, it is impossible to have an up-to-date national picture at any one point in time.

⇒ Very few formularies referred to specific diseases or populations (age and ethnicity), so we were unable report recommendations at this level.

⇒ Ambiguity in many of the formularies, differences in researcher interpretation and transcription errors may mean inaccuracies were introduced during the data extraction process.

responsibility for the provision of National Health Service (NHS) care in their locality, maintain local prescribing formularies. National Institute for Health and Care Excellence (NICE) guidelines states that such information should be published '… in a clear, simple and transparent way, so that patients, the public and stakeholders can easily understand it'.[5]

The need for multiple formularies is questionable, although the number, type and content of recommendations made has never been examined. Therefore, we sought to identify and compare the structure and content of emollient formularies across England and Wales, including any rationale behind the recommendations given.

## INTRODUCTION

The annual spend on emollients in England alone in 2015/2016 was £116.2 million.[1] Despite their accepted value in dry skin conditions such as atopic eczema, there are multiple products that come in different formulations (ointments, gels, creams, lotions, sprays and bath additives) and a weak evidence base to guide their use.[2–4]

Most prescribing of emollients happens in primary care and clinical commissioning groups (CCGs) in England and local health boards (LHBs) in Wales, who have the

## METHODS
### Data collection

First, CCGs in England and LHBs in Wales were identified using the NHS choices (www.nhs.uk), NHS England (www.england.nhs.uk, as listed in October 2016) and NHS Wales websites (www.wales.nhs.uk, February 2017). All available formularies were identified by systematically searching each CCG/LHB

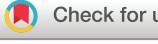

Population Health Sciences, Bristol Medical School, University of Bristol, Bristol, UK

**Correspondence to**
Dr Matthew J Ridd;
m.ridd@bristol.ac.uk

website or undertaking Google searches using relevant keywords.

JPC and GB extracted data from each formulary into 'leave on' and bath additive Excel spreadsheets templates, respectively. These templates were initially piloted and developed with a sample of formularies. To ensure consistency of data extraction, a codebook detailing the various coding options was developed alongside. An independent check of 10 formularies identified discrepancies in 4.7% (57/1220) of 'leave on' emollient (PAQ) and 9.8% (44/448) of bath additive (JPC) data points. Where disagreements arose, the two data extractors discussed and agreed what should be recorded; most stemmed from differing interpretations of vague and/or inconsistent formularies rather than random errors.

## Analysis

Data were then imported and analysed descriptively using STATA (V.14). All percentages were rounded to the nearest whole number unless stated otherwise.

## Patient involvement

'Which emollient is the most effective and safe in treating eczema?' was identified as a priority for research in a patient and clinician-led prioritisation exercise published in 2013.[6] Public and patient involvement in subsequent eczema research led by MJR has reinforced the importance of understanding the variation in emollient prescribing and recommendations made by different doctors and specialist nurses better.[7–9] However, patients were not involved in the design of the study or the interpretation of the findings.

## RESULTS

Eighty-nine per cent (185/209) of CCGs had an emollient formulary and all seven LHBs had an emollient formulary (figure 1). Seventy-one per cent (131/185) CCGs shared an emollient formulary with at least one other CCG, but all seven LHBs had their own individual emollient formulary. Some CCGs (24/209) withheld formularies for internal use only or did not respond, even after direct correspondence, meaning data could only be collected from 89% of CCGs. Therefore, overall 102 formularies were examined.

The structure of formularies varied, but for emollients, most (63%, 64/102) adopted a 'rank' (traffic light, medal or number) structure, with by formulation (51%, 52/102) being next most common, followed by basic (alphabetical) list (47%, 48/102) and by skin dryness (10%, 10/102) (categories not mutually exclusive). Thirty-four per cent (35/102) of formularies gave the specific costs for each emollient.

## Emollients

Over 109 different emollients were named, with 93 of these recommended by at least one formulary. Almost every formulary recommended at least one cream (99%)

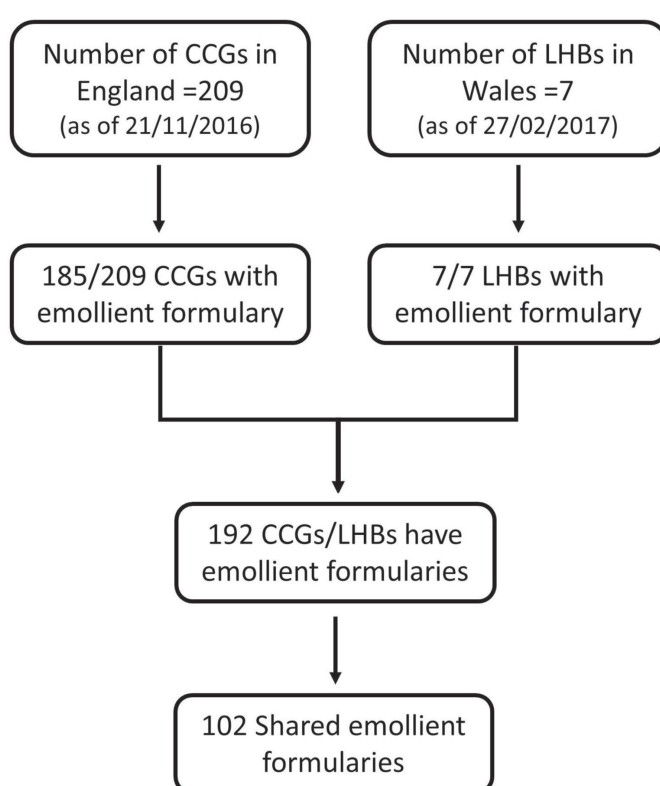

**Figure 1** Flow chart of the clinical commissioning groups (CCGs)/local health boards (LHBs) in England/Wales and number of available emollient formularies (see accompanying document).

or ointment (98%) (see table 1). Most formularies also recommended at least one lotion or gel (both 85%), but only 29% recommended a spray. No formularies censured the use of creams, lotions, ointments or gels. However, 9% of formularies recommended against sprays and most formularies (62%) made no mention of them. 'Other' types of emollients were most commonly (58%) not recommend by formularies. This category contained more unusual products such as food-based oils and balms (online supplementary table S1).

Cream and ointments were the most commonly recommended types of emollients (table 2). Of the specifically recommended and named emollients, three of the 'top five' are ointments (table 2). However, between 2% and 9% of formularies specifically did not recommend one or more of these products. When ranked by emollient type (lotion, cream, gel and ointment), three products that contain urea or antimicrobials (Dermol 500 Lotion, Dermol Cream and Balneum plus Cream) appear in the 'top five' (online supplementary table S2). Among the 64 formularies that specified an order of preference for recommended emollients, generic emulsifying ointment was the most popular first-line emollient, followed by three 'Zero' products and white soft/liquid paraffin 50/50 (online supplementary table S3). Aqueous cream was the most common emollient not recommended (45% of formularies, table 2), although again there was disagreement between formularies, with up to 16%

**Table 1** Formularies' recommendations by type of emollient (n=102)

| Formulation | Recommended | | Not recommended | | Neither recommended nor not recommended | |
|---|---|---|---|---|---|---|
| | n | % | n | % | n | % |
| Lotion | 87 | 85 | 0 | 0 | 15 | 15 |
| Cream | 101 | 99 | 0 | 0 | 1 | 1 |
| Gel | 87 | 85 | 0 | 0 | 15 | 15 |
| Ointment | 100 | 98 | 0 | 0 | 2 | 2 |
| Spray | 30 | 29 | 9 | 9 | 63 | 62 |
| Other | 43 | 42 | 59 | 58 | 0 | 0 |

recommending one or more of the top five not recommended emollients.

The most common reasons given for the recommendations made were patient preference (60%, 61/102) and/or product cost (58%, 59/102). Forty-two per cent (43/102) of formularies gave other reasons, the most common being 'the greasier the emollient, the better the effect'. Other rationale given for not recommending emollients included concerns over food-based ingredients and the ability of active ingredients to penetrate the skin. No rationale was given in 28% (29/102).

### Bath additives

Of the 82% (84/102) of formularies that recommended the use of bath additives, 75% (64/84) did not give any rationale. Six formularies noted that evidence to support their use was lacking, eight recommended their use only in specific circumstances and six cited 'possible benefit for some patients'.

There was no mention of bath additives in 7% (7/102) of formularies and 11% (11/102) did not recommend their routine use. Four per cent (4/102) stated that patients should buy their own. The rationale for not prescribing bath emollients was variable, the most common (non-exclusive) reasons cited were: lack of evidence (5/11), concerns about safety (4/11), the potential to undertreat eczema by distracting from leave-on emollient use (3/11) and cost (1/11). Three formularies did not give a reason.

Most formularies (92%, 77/84) presented their recommendations as individually named bath additives. A small number of formularies (8%, 7/84) also suggested that any emollient (except for '50:50') could be used in the bath instead of a specific bath additive. Regarding the use of bath additives with antimicrobials, they were recommended in 74% (62/84) of formularies, not mentioned in 25% (21/84), and one advised against their use ever. Of those recommending their use, 34% (21/62) specified 'only in the presence of infection'.

Overall, 19 regular and five antimicrobial containing bath additives were named (see table 3). The five most commonly recommended bath emollients were Oilatum, Hydromol, Dermol 600, Oilatum plus and Balneum plus. Some formularies restricted use of specific bath additives;

**Table 2** Top five emollients most commonly recommended and not recommended by formularies (all types) (n=102)

| Name of emollient | Number (%) of formularies | | | |
|---|---|---|---|---|
| | Recommending | | Not recommending | |
| | n | % | n | % |
| **Top five recommended** | | | | |
| White Soft/Liquid Paraffin 50/50 | 81 | 79 | 2 | 2 |
| Emulsifying Ointment BP | 80 | 78 | 2 | 2 |
| Hydromol ointment | 72 | 71 | 6 | 6 |
| Dermol 500 Lotion | 71 | 70 | 9 | 9 |
| Cetraben Cream | 70 | 69 | 6 | 6 |
| **Top five not recommended** | | | | |
| Aqueous Cream | 11 | 11 | 46 | 45 |
| E45 itch relief | 10 | 10 | 22 | 22 |
| Ultrabase Cream | 16 | 16 | 20 | 20 |
| Nutraplus Cream | 14 | 14 | 19 | 19 |
| Dermamist Spray | 12 | 12 | 18 | 18 |
| Lipobase | 3 | 3 | 18 | 18 |

**Table 3** Regular and antimicrobial containing bath emollients recommended by formularies (n=84)

| | Recommended | | Not recommended | |
|---|---|---|---|---|
| | n | % | n | % |
| **Regular bath emollients** | | | | |
| Oilatum | 51 | 61 | 14 | 17 |
| Hydromol | 43 | 51 | 10 | 12 |
| Balneum plus | 32 | 38 | 9 | 11 |
| Balneum | 30 | 36 | 17 | 20 |
| Aveeno | 24 | 29 | 20 | 24 |
| Zeroneum | 24 | 29 | 20 | 24 |
| Dermalo | 22 | 26 | 20 | 24 |
| Zerolatum | 20 | 24 | 21 | 25 |
| Double base | 19 | 23 | 12 | 14 |
| Cetraben | 18 | 21 | 16 | 19 |
| QV bath oil | 15 | 18 | 20 | 24 |
| LPL 63.4 | 13 | 15 | 10 | 12 |
| E45 | 13 | 15 | 21 | 25 |
| Aqueous cream BP | 11 | 13 | 9 | 11 |
| Diprobath | 10 | 12 | 18 | 21 |
| Oilatum Junior | 8 | 10 | 9 | 11 |
| Zerolatum plus | 4 | 5 | 16 | 19 |
| ZeroAQS | 2 | 2 | 7 | 8 |
| Zerozole | 1 | 1 | 18 | 21 |
| **Bath emollients containing antimicrobial agent** | | | | |
| Dermol 600 | 41 | 49 | 8 | 10 |
| Oilatum plus | 34 | 40 | 11 | 13 |
| Emulsiderm | 22 | 26 | 18 | 21 |
| Dermol | 14 | 17 | 14 | 17 |
| Dermol 500 | 4 | 5 | 7 | 8 |

for example, Aveeno was often only recommended if paraffin intolerant or following no response to other bath emollients.

## DISCUSSION

We found that most CCGs and LHBs (192/216) had a formulary, with 102 unique emollient formularies between them. Of the 109 different emollients named, creams and ointments were the most commonly recommended types, and three ointments (White Soft/Liquid Paraffin 50/50, Emulsifying Ointment and Hydromol Ointment) were the most frequently specified products. However, there was poor consensus over which emollient should be used first line, and 4 out of 10 of the most recommended lotions and cream contain antimicrobials or urea. Patient preference (60%) and/or cost (58%) were the most common reasons given for the recommendations. However, cost recommendations were on a 'price

per gram or millilitre' basis, rather than proper cost-effectiveness evidence. Eleven per cent of formularies did not recommend the use of bath additives, while the 82% that did named 24 different bath additives and most (75%) did not give any reasons for their recommendation.

This is the first study to compare all emollient formularies in England and Wales. Formularies were identified, and data were extracted in a systematic way. Ambiguity in many of the formularies, differences in interpretation and transcription errors may mean inaccuracies were introduced during this process. In the time between collecting and reporting the data, two further CCGs have merged (new total 207), and a rapid review of the emollient formularies in November 2017 identified that 50% have been updated. However, most changes appear to be minor, and the key findings of the number of formularies and similarities/differences between the different formularies hold true. Because of the labour-intensive nature of data extraction, the constant change in NHS service configuration, and CCGs/LHBs independently maintaining >100 formularies, it is impossible to have an up-to-date national picture at any one point in time. Very few formularies referred to specific diseases or populations (age, ethnicity), so we were unable report recommendations at this level.

One significant factor contributing to the lack of agreement between the formularies is the weak evidence base. Several recent systematic reviews have all concluded that good quality research comparing the effectiveness or acceptability of emollients is very limited.[2–4] Likewise, there is an absence of evidence to demonstrate the clinical and cost-effectiveness of bath additives.[3] The greatest consensus is around recommendations for ointments, one appeal being their low cost, although current NICE guidance on atopic eczema in children (from 2007) notes that they are often less acceptable and are therefore 'not usually suitable as a first-line treatment'.[10] The predominance of emollients that contain urea or antimicrobials is surprising: these products are usually second-line treatments, generally cost more than standard emollients and are more likely to irritate skin.[1 11] It is encouraging to see 45% of formularies actively not recommending aqueous cream, probably reflecting the 2013 Medicines and Healthcare products Regulatory Agency (MHRA) safety report on adverse skin reactions in children with eczema.[12] There is a disparity between formulary recommendations and recent NHS community dispensing data (for children and adults)[13 14] where the top five dispensed emollients were Cetraben Cream, Doublebase gel, Diprobase cream, Aveeno cream and Dermol 500 Lotion. There may be many reasons for this observation, beyond the scope of this paper, but which may include: a time lag between formulary implementation and prescribing patterns changing; prescriber lack of familiarity with formulary recommendations; and/or strong clinician or patient preference for non-recommended emollients.

We know that much emollient prescribing is done on a 'trial and error' basis,[15 16] but the extent to which all these

different formularies help reduce (or indeed contribute) to this problem is unclear. While local formularies can reflect the needs of the population served, the justification in the case of emollients/bath additives is weak (and indeed local factors contributing to decision making not specified in any of the formularies). Furthermore, we cannot understand why the formularies of 24 CCGs are not publicly available. In addition to the time and cost of different medicine management teams and area prescribing committees developing individual emollient formularies, differences between them serves to cause confusion to prescribers and patients, especially when they move between areas. This combined with the discrepancy between emollient dispensing data and formulary recommendations suggests the purpose of emollient formularies should be rethought and/or communicated more effectively.

While the 'correct' emollient is the 'one that the child will use'[10] is a pragmatic approach to emollient prescribing, it is not one that is supported by formulary recommendations that have a weak evidence base. Future research should further explore the disparity between recommended and observed prescribing of emollients, including changes over time in formulary recommendations and prescribing patterns. The results of ongoing pragmatic trials such as Best Emollient for Eczema (comparing different emollient types, https://doi.org/10.1186/ISRCTN84540529) and Bath Additives for the Treatment of Childhood Eczema (bath additives in addition to usual care)[17] will provide clinical and cost-effectiveness data that will inform, and hopefully simplify, formulary recommendations to the benefit of patients and clinicians.

**Contributors** MJR conceived the study. JPC and GB undertook data extraction and analyses (for emollients and bath additives, respectively), supervised by MJR. Independent data scrutiny was done by PAQ (emollients) and JPC (bath additives). JPC, GB and MJR interpreted the data. JPC drafted the manuscript and MJR substantially revised it, with contributions from GB and PAQ. All authors had full access to all the data in the study and take responsibility for the integrity of the data and the accuracy of the data analysis. MJR is guarantor for the study.

**Funding** MJR was supported by the National Institute for Health Research during this study (NIHR Post Doctoral Fellowship, PDF-2014-07-013). JPC and GB received funding from the Academy of Medical Sciences INSPIRE scheme, University of Bristol.

**Disclaimer** The views expressed in this article are those of the author(s) and not necessarily those of the NHS, the NIHR or the Department of Health. Funders of this study had no role in the study design; collection, analysis and interpretation of the data; writing of the manuscript; or decision to submit the manuscript for publication. The authors are independent from funders.

**Competing interests** MJR reports grants from National Institute for Health Research (NIHR) Post Doctoral Fellowship (PDF-2014-07-013) and NIHR HTA (11/153/01 and 15/130/07). JPC and GB report grants from The Academy of Medical Sciences INSPIRE scheme.

**Patient consent** Not required.

**Ethics approval** This study did not require ethics approval.

**Provenance and peer review** Not commissioned; externally peer reviewed.

**Data sharing statement** Full dataset and technical appendix have been uploaded as supplementary files.

**ORCID iD**
Matthew J Ridd http://orcid.org/0000-0002-7954-8823

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
