## [Reviewer comments · BMJ Open]

ARTICLE DETAILS

TITLE (PROVISIONAL)	Emollient prescribing formularies in England and Wales: cross-sectional study
AUTHORS	Chan, Jonathan; Boyd, Grace; Quinn, Patrick; Ridd, Matthew

VERSION 1 – REVIEW

REVIEWER	S B Ang KK Women's and Children's Hospital, Singapore
REVIEW RETURNED	08-Feb-2018

GENERAL COMMENTS	This is a useful overview of practices in terms of emollient use and the investigators have put in a lot of effort in view of the large datasets they are likely to have to work with. The following are some comments that can improve the study. Please explain how you derive the recommendation for or against from just extracting data from the formulary. The methodology needs more clarity on how some of the results were derived. I am unable to figure out how the results were derived based on the methodology described. Please find evidence to support this statement made in the discussion: The predominance of emollients which contain urea or antimicrobials is surprising: these products are usually second-line treatments, generally cost more than standard emollients, and are more likely to irritate skin. I find the comment "This may be due to a lag between formulary implementation and prescribing patterns changing, wilful ignorance of formularies on the part of prescribers or strong patient preference for alternative emollients" a little too simplistic. Perhaps you could examine the available evidence that might have supported this practice rather than putting a blanket statement that the prescribers are doing something that is wrong.
--

REVIEWER	Dr Lisa Szatkowski University of Nottingham, UK
REVIEW RETURNED	12-Feb-2018

GENERAL COMMENTS	Thank you for the request to review this article. The work is clear and concise and generally well-written. I have the following specific minor recommendations: - In order to allow the study to be replicated, please consider
---

	publishing the data extraction template and codebook as supplementary information alongside the article  - Please describe in your methods how you dealt with discrepancies in data extraction. - Phrasing could be tightened up in the text and the table column headings to make clear the difference between formularies ACTIVELY not recommending the use of particular products and there being simply no mention of them - The last paragraph of the discussion focusses on emollient use in children. Is anything known about the relative volume of prescribing of emollients to children and adults? If so, it would be useful to briefly describe this in the introduction. If you set out with the initial aim of describing variations in formularies by factors such as age, please state this in your methods. - Please clarify whether the data on dispensed emollients in England (p6 line 24) relate to children specifically, or the whole population?
--	--

VERSION 1 – AUTHOR RESPONSE

Reviewer 1

1. Please explain how you derive the recommendation for or against from just extracting data from the formulary. The methodology needs more clarity on how some of the results were derived. I am unable to figure out how the results were derived based on the methodology described.

Thank you for seeking clarification on this matter. Only emollients that were named and explicitly listed on each formulary as being recommended/not recommended were recorded as such during data extraction and analysis. Where there was ambiguity, we developed and applied consistent rules as detailed in our codebook, which we have uploaded as supplementary material for publication.

2. Please find evidence to support this statement made in the discussion: The predominance of emollients which contain urea or antimicrobials is surprising: these products are usually second-line treatments, generally cost more than standard emollients, and are more likely to irritate skin.

We have now added two references to support this statement (Ling & Highet, Journal of Dermatological Treatment 2000; 11: 263; and Wilson & Smith PrescQIPP 2015).

3. I find the comment "This may be due to a lag between formulary implementation and prescribing patterns changing, wilful ignorance of formularies on the part of prescribers or strong patient preference for alternative emollients" a little too simplistic. Perhaps you could examine the available evidence that might have supported this practice rather than putting a blanket statement that the prescribers are doing something that is wrong.

On reflection, we recognise that we may have over-stated our thoughts on this matter, so we have revised this sentence as follows: "There may be many reasons for this observation, beyond the scope of this paper, but which may include: a time lag between formulary implementation and prescribing patterns changing; prescriber lack of familiarity with formulary recommendations; and/or strong clinician or patient preference for non-recommended emollients."

Reviewer 2

1. In order to allow the study to be replicated, please consider publishing the data extraction template and codebook as supplementary information alongside the article.

Thank you for this suggestion. In response to this and comment 1 from review 1 (see above), we have uploaded both the data extraction template and codebook as supplementary material.

2. Please describe in your methods how you dealt with discrepancies in data extraction.

Some error or disagreement between individuals carrying out data extraction of this nature is inevitable given the number of emollients and bath additives, formularies, and the variation in their layout and content. Given these considerations, we feel that over 90% agreement across 'leave on' emollient and bath emollient formularies is acceptable. Where discrepancies were identified during the checking exercise, the two extractors discussed and agreed what should be recorded. The final dataset used for the analyses included any these changes.

Accordingly, we have added the following sentence to the methods section: "Where disagreements arose, the two data extractors discussed and agreed what should be recorded; most stemmed from differing interpretations of vague and/or inconsistent formularies rather than random errors."

3. Phrasing could be tightened up in the text and the table column headings to make clear the difference between formularies ACTIVELY not recommending the use of particular products and there being simply no mention of them

Thank you for this suggestion. Text and headings have been altered through the manuscript to make the distinction clearer.

4. The last paragraph of the discussion focusses on emollient use in children. Is anything known about the relative volume of prescribing of emollients to children and adults? If so, it would be useful to briefly describe this in the introduction. If you set out with the initial aim of describing variations in formularies by factors such as age, please state this in your methods.

We are not aware of any data on the relative volumes of emollients prescribed between adults and children

With regards to our aims, it became quickly apparent when we initially "scoped" the formularies as part of developing the data extraction tool and codebook, that most formularies did not make recommendations by age or diagnosis, so we did not collect this data on the occasions when it was specified.

5. Please clarify whether the data on dispensed emollients in England (p6 line 24) relate to children specifically, or the whole population?

The data on dispensed emollients was not specifically related to children but was population wide data in the community (primary care) across England and Wales. We have added this detail to the relevant sentence for clarity: "There is a disparity between formulary recommendations and recent NHS community dispensing data (for children and adults), ..."

VERSION 2 – REVIEW

REVIEWER	Dr Lisa Szatkowski University of Nottingham, UK
REVIEW RETURNED	26-Mar-2018

GENERAL COMMENTS	I am happy to recommend this article for acceptance
---

bmjopen-2018-022009.R